# Endorsement of the 32-item Consolidated Criteria for Reporting Qualitative (COREQ) and Standards for Reporting Qualitative Research (SRQR) by Chinese journals of nursing: A survey of editors and review of journal instructions for authors

**Li Zhao**[1,2,3☯], **Jian Zeng**[4☯], **Kaiyan Hu**[3☯], **Bin Ma**[3,5]*

**1** Beijing University of Chinese Medicine Shenzhen Hospital (Longgang), Shenzhen, China, **2** Shenzhen Health Capacity Building and Continuing Education Center, Shenzhen, China, **3** Evidence Based Medicine Center, School of Basic Medical Sciences, Lanzhou University, Lanzhou, China, **4** Department of Anesthesiology, Longgang District Central Hospital of Shenzhen, Shenzhen, China, **5** Lanzhou University, Key Laboratory of Evidence-Based Medicine and Knowledge Translation of Gansu Province, Lanzhou, China

☯ These authors contributed equally to this work.
* mabin1909@outlook.com

## Abstract

### Background

To investigate the endorsement of the 32-item Consolidated Criteria for Reporting Qualitative (COREQ) and Standards for Reporting Qualitative Research (SRQR) in the instructions for authors (IFA) of Chinese nursing journals. The awareness of Chinese editors of the COREQ and SRQR, together with their application and requirements for following the standards, were also investigated. These findings would assist as the endorsement, application, and promotion of the COREQ and SRQR in Chinese nursing journals, and improve the reporting quality of qualitative research in nursing.

### Methods

Nursing journals were identified from the National Press and Publication Administration. The IFA and applications of the COREQ and SRQR were assessed. The editors of the journals were asked about their awareness of and demand for the COREQ checklist and SRQR standards, as well as their implementation at different stages of the publication process, including manuscript submission, editing, and peer review.

### Results

A total of 29 nursing journals were included, and only 2 journals (6.9%, 2/29) mentioned the COREQ and SRQR in their IFA. Among the 24 surveyed editors, only 45.83% (11/24) and 33.33% (8/24) were aware of the COREQ and SRQR,

**Data availability statement:** All relevant data are within the paper and its Supporting Information files.

**Funding:** The author(s) received no specific funding for this work.

**Competing interests:** The authors have declared that no competing interests exist.

respectively. None of the surveyed editors required authors to follow the COREQ/SRQR at the submission stage, editors to follow COREQ/SRQR in the journal editing and processing stage, and reviewers to use the COREQ/SRQR in the expert review stage.

## Conclusion

Nursing journals in China endorsing the COREQ and SRQR constitute a small percentage of the total. In addition, both awareness and application of the COREQ and SRQR were poor among nursing journal editors. Therefore, we strongly recommend that the China Periodicals Association undertake measures to encourage and support the endorsement of biomedical research reporting guidelines in nursing journals. Also, the education and training of journal editors, researchers, and medical students on biomedical research reporting guidelines should be strengthened.

## Introduction

Qualitative research explores concepts that are difficult to quantify to elucidate their meanings and understand the complex phenomena and processes behind patterns of behavior in natural settings [1]. Compared with quantitative research, qualitative research methods can be used to explore people's values, experiences, preferences, behaviors, and feelings, which can compensate for deficiencies in quantitative research arising from a lack of focus on the experiences of research subjects. Qualitative methods are also useful for solving specific problems that are otherwise difficult to solve using quantitative methods [2].

Qualitative research methods can not only be used independently but also in conjunction with quantitative methods to obtain a more comprehensive assessment of the experiences and effects of treatment, and are useful in situations such as preliminary assessments of the impact of interventions before the commencement of clinical trials, and analyzing the reasons for subjects' withdrawal from trials or refusing treatment during a trial [3,4]. Therefore, qualitative research plays an important role in nursing and cannot be ignored in the context of biopsychosocial medicine and holistic care. Since the results of qualitative research are in the form of textual narratives rather than in a quantitative form, readers can only understand research methods, their implementation, and analyses of results to the extent that the textual content is reported by the authors [5,6]. Hence, inadequate reporting of qualitative studies may prevent readers' assessment of the rigor and reliability of the methods, and may also reduce the credibility and application of the results, mislead readers, and adversely affect the development of future studies [7,8].

To address these issues and enhance transparency in the reporting of qualitative research and avoid the potential compromise of high-quality studies in credibility assessments due to incomplete reporting, the 32-item Consolidated Criteria for Reporting Qualitative (COREQ) and Standards for Reporting Qualitative Research (SRQR) were published in 2007 and 2014, respectively, to standardize the reporting

of qualitative studies [7,9]. In recent years, reporting guidelines for different types of biomedical research have been developed for standardization [10–12]. Studies have shown that the introduction of clinical research reporting guidelines such as the Consolidated Standards of Reporting Trials (CONSORT) and Strengthening the Reporting of Observational Studies in Epidemiology (STROBE) guidelines in the instructions for authors (IFA) of journal publications can effectively improve the reporting quality of biomedical research [13,14]. Both COREQ and SRQR have been endorsed and recommended by the EQUATOR (enhancing the quality and transparency of health research) website, and the COREQ checklist has been introduced into the IFA by several journals included in the Science Citation Index (SCI) [15,16]. However, Walsh et al. recently found that only 28% of the 71 nursing journals surveyed mentioned the COREQ checklist in their IFA and that the reporting quality of qualitative studies published in these 71 nursing journals was inadequate [17].

In recent years, qualitative research has been increasingly used in the nursing field in China, especially after 2007, and the number of articles published has increased almost linearly with time [18]. However, no research has as yet investigated the endorsement of the COREQ and SRQR in the IFA of Chinese nursing journals. In addition, since journal editors are directly responsible for determining the reporting format of submitted manuscripts, they play an important role in promoting author adherence to reporting guidelines and in promoting the use of reporting guidelines by editors and reviewers [19]. A study by Loscalzo has shown that active implementation of the CONSORT guidelines by journal editors can improve the reporting quality of abstracts, but there is no research on the awareness and implementation of the COREQ and SRQR by journal editors in the field of nursing in China [20].

Therefore, this study first adopted the method of retrospective analysis to comprehensively understand the endorsement of the COREQ and SRQR in the IFA of Chinese nursing journals. Then, we used questionnaires to evaluate editors' awareness of COREQ and SRQR and their demand for and implementation of the guidelines at different stages of the publication process, such as manuscript submission, editing, and peer review. The aim was to promote the introduction, application, and widespread use of the COREQ and SRQR in Chinese nursing journals, and to provide a scientific reference and guide for improving the quality of qualitative research reports in nursing.

## Materials and methods

### Journals

The National Press and Publication Administration (NPPA) serves as the supreme administrative authority governing journal registration and management in mainland China. Its registry of officially sanctioned publications constitutes the definitive legal basis for verifying a journal's legitimate publication status within this jurisdiction. Accordingly, we extracted the complete nomenclature of nursing journals holding valid registration credentials from the NPPA official portal to establish our initial journal pool.Complementarily, we interrogated the Chinese Science Citation Database (CSCD) and the Chinese Scientific and Technical Papers and Citations Database (CSTPCD) to identify high-impact journals within them. Both databases encompass preeminent journals across diverse disciplinary domains in China; notably, the CSCD is widely recognized as the Chinese equivalent of the Science Citation Index.

Inclusion criteria were as follows: (i) journals holding valid registration and legal publication status in mainland China; (ii) journals explicitly categorized within the nursing discipline; and (iii) journals with active, regular publication schedules. Exclusion criteria comprised: (i) journals with no record of qualitative research publications; and (ii) journals published in the Hong Kong Special Administrative Region, the Macao Special Administrative Region, or Taiwan, China.

We excluded nursing journals published in the Hong Kong SAR, the Macao SAR, and Taiwan, China. Although these regions share a common Chinese cultural heritage, their academic publishing ecosystems—including journal evaluation frameworks, editorial policies, and peer review standards—differ systemically from those of mainland China. This exclusion was deliberately imposed to ensure sample homogeneity with respect to policy and practice contexts, thereby capturing the specific landscape shaped by unified administrative governance and dominant academic culture within mainland China.

### Extraction of instructions for authors

The latest versions of the IFA (dated May 2023 or earlier) of each included journal were downloaded from their official website. Two researchers (LZ and KH) independently read the IFA, extracted any information regarding the COREQ and SRQR, and cross-checked the extracted results. If there were any disagreements, they were resolved through discussion or adjudicated by a third party (BM).

### Investigation of periodical editors

We obtained contact information such as phone numbers and email addresses of the editorial department or relevant editors of each included journal by visiting the official websites of the journals. Two investigators (LZ and KH) first sent questionnaires to the editorial department or relevant editors of the journals by email, with a total of three rounds of e-mail, at two-week intervals. For those journals that did not respond after three rounds of emails, investigators conducted one-on-one telephone interviews.

The questionnaire was designed by the research group based on literature review and research purpose. The question types included both single-choice and multiple-choice questions, and open-ended questions. The main questions were as follows: 1) basic information of the respondents; 2) respondents' awareness of the COREQ; 3) respondents' awareness of the SRQR; 4) application status of the COREQ and SRQR during the process of manuscript submission by authors, review by editors, and peer review by reviewers; 5) obstacles blocking the introduction of the COREQ or SRQR into the routine processes of journals and the determining factors for Chinese journals to adopt the COREQ or SRQR. The survey was conducted between July 2023 and November 2023.

### Research quality control

In this study, all investigators were trained before the formal questionnaire survey. The training content included content specific to the COREQ and SRQR, the meaning of each item in the questionnaire, requirements for completing the questionnaire, communication style, tone, skills, and question consultation methods during the telephone survey, to ensure the authenticity and credibility of the formal survey results.

### Ethical statement

This study was approved by the Ethics Committee of the School of Nursing, Lanzhou University(Approval Number:LZUHLXY20210099). The content of this study was deemed to be of no harm to the participants, so we obtained consent from the participants by e-mail or phone.

### Statistical analysis

Excel® (version Microsoft Excel 2016; http://office.microsoft.com/zh-cn/) software was used to analyze the data. The small sample ($N < 30$) compromises statistical stability and elevates Type I/II error risk [21,22], furthermore, the study focus was on description, not hypothesis testing. Therefore, descriptive statistics were applied, with categorical data reported in counts ($n$) and percentages (%).

### Patient and public involvement statemen

Patients or the public were not involved in the design, conduct, reporting, or dissemination plans of this research. This was due to resource constraints. Findings will be shared with public communities through open-access publication.

## Results

### Number and types of included journals

A total of 29 nursing journals were included in this study, eight of which were included in the CSCD and six in the CST-PCD. None were indexed in MEDLINE. The complete journal list is provided in S1 File.

**Addition of the COREQ/SRQR to the IFA of Chinese nursing journals**

Among the 29 nursing journals included in this study, only two journals (2/29, 6.9%) that were included in the CSCD database mentioned the COREQ/SRQR in their IFA. Of these two journals, one was the *International Journal of Nursing Sciences*, which provides the reference website for the COREQ in its IFA. The other journal was the *Chinese Nursing Management*, which provides the interpretation article of the SRQR in its IFA.

**Awareness, demand, and application status of the COREQ/SRQR among editors of Chinese nursing journals**

The investigators approached the editors of 29 Chinese nursing journals and received 24 valid questionnaires with a response rate of 82.76%. The respondents included 18 editors (75%, 18/24), 3 editorial directors (12.5%, 3/24), and 3 intern editors (12.5%, 3/24).

**Awareness and demand status of the COREQ/SRQR among editors of Chinese nursing journals (Table 1)**

Among the 24 editors who completed the survey, only 11 (45.83%, 11/24) editors were aware of the COREQ, while only 9.09% (1/11) of them were fully familiar with its contents. The survey also showed that only 8 (33.33%, 8/24) editors were aware of the SRQR, while only 12.5% (1/8) of them were fully familiar with its contents. The editors had learned about the COREQ/SRQR from reading the literature (58.33%, 7/12), academic reports (33.33%, 4/12), and network resources (33.33%, 4/12).

Of the 12 editors (50%, 12/24) who knew about the COREQ or the SRQR or both, 91.67% (11/12) believed it was necessary to add the COREQ and/or SRQR to the IFA of nursing journals, and 83.33% (10/12) believed it was necessary to follow the COREQ/SRQR guidelines during the processes of manuscript submission, editing, and peer review. However, of the 12 editors (50%, 12/24) who did not know of either COREQ or SRQR, only 50% (6/12) believed it was necessary to add the COREQ and/or SRQR to the IFA of nursing journals, and fewer than half thought it was necessary to ask authors to follow the COREQ/SRQR at the manuscript submission stage (41.67%, 5/12), ask editors follow them during the manuscript processing and editing stages (41.67%, 5/12), and ask reviewers use them at the expert review stage (33.33%, 4/12).

Six of the 12 editors (50%, 6/12) who were not aware of both the COREQ and the SRQR were willing to learn more about the COREQ/SRQR. Among them, reading the literature (41.66%, 5/12), online courses (41.66%, 5/12), and offline training (41.66%, 5/12) were the most acceptable learning avenues.

**Application status of the COREQ/SRQR among editors of Chinese nursing journals (Table 2)**

The survey revealed that none of the respondents had, in fact, strongly requested authors to follow the COREQ/SRQR at the manuscript submission stage, the editors to follow them at the manuscript processing and editing stages, and the reviewers to use them at the expert review stage. Only about one-third of the respondents recommended that authors follow the COREQ/SRQR at the manuscript submission stage (33.33%, 8/24), that editors follow them at the manuscript processing and editing stages (29.17%, 7/24), and that reviewers use them at the expert review stage (33.33%, 8/24). In addition, 68.18% (15/22) of the editors of journals that had not added the COREQ/SRQR in their IFA thought it was necessary to introduce the COREQ/SRQR into the IFA of nursing journals. However, only 13.33% (2/15) of them intended to introduce the COREQ/SRQR into the IFA of their journals in the near future, and none of them had a specific plan to do so.

**Reasons for not introducing the COREQ/SRQR into the IFA of Chinese nursing journals (Table 3)**

The 24 editors who completed the survey considered that the main determinants for introducing the COREQ/SRQR in the IFA of Chinese nursing journals were as follows: formal approval from the supervisory departments in charge (66.67%, 16/24); proposal and request by the editor-in-chief (58.33%, 14/24); consideration of international practices

**Table 1. Awareness and demand status of the COREQ/SRQR among nursing journal editors in China.**

| Journal editors' awareness and demand for the COREQ/SRQR | Number (%) of N = 24 |
|---|---|
| 1. Editors who were aware of the COREQ/SRQR | 12(50%,12/24) |
| 1.1 Editors who were aware of the COREQ | 11(45.83%,11/24) |
| 1.1.1 The extent to which editors were aware of the COREQ | |
| Did not understand the core content of the COREQ | 1(9.09%,1/11) |
| Understood part of the content of the COREQ | 9(81.82%,9/11) |
| Were familiar with the entire content of the COREQ | 1(9.09%,1/11) |
| 1.2 Editors who were aware of the SRQR | 8(33.33%,8/24) |
| 1.2.1 The extent to which editors were aware of the SRQR | |
| Did not understand the core content of the SRQR | 2(25%,2/8) |
| Understood part of the content of the SRQR | 5(62.5%,5/8) |
| Were familiar with the entire content of the SRQR | 1(12.5%,1/8) |
| 1.3 Ways of acquiring information on the COREQ/SRQR | |
| Literature reading | 7(58.33%,7/12) |
| Academic reports | 4(33.33%,4/12) |
| Network resources | 4(33.33%,4/12) |
| Short-term training | 2(16.67%,2/12) |
| Professional books | 2(16.67%,2/12) |
| Other | 2(16.67%,2/12) |
| 1.4 Editors who thought it necessary to introduce the COREQ/SRQR into the IFA | 11(91.67%,11/12) |
| 1.5 Editors who thought it necessary to require authors to comply with the COREQ/SRQR | 10(83.33%,10/12) |
| 1.6 Editors who thought it necessary to require editors to incorporate COREQ/SRQR into editorial processes | 10(83.33%,10/12) |
| 1.7 Editors who thought it necessary to require peer reviewers to incorporate COREQ/SRQR into peer review processes | 10(83.33%,10/12) |
| 2. Editors who were not aware of the COREQ/SRQR | 12(50%,12/24) |
| 2.1 Willing to further study and understand the COREQ/SRQR in the future | 6(50%,6/12) |
| 2.1.1 Ways to learn about the COREQ/SRQR in the future | |
| Literature reading | 5(41.66%,5/12) |
| Professional books | 3(25%,3/12) |
| Online courses | 5(41.66%,5/12) |
| Offline training | 5(41.66%,5/12) |
| Instructions for authors | 3(25%,3/12) |
| Other | 0 |
| 2.2 Editors who thought it necessary to introduce the COREQ/SRQR into the IFA | 6(50%,6/12) |
| 2.3 Editors who thought it necessary to require authors to comply with the COREQ/SRQR | 5(41.67%,5/12) |
| 2.4 Editors who thought it necessary to require editors to incorporate the COREQ/SRQR into editorial processes | 5(41.67%,5/12) |
| 2.5 Editors who thought it necessary to require peer reviewers to incorporate the COREQ/SRQR into peer review processes | 4(33.33%,4/12) |

**Table 2. Application status of the COREQ/SRQR among editors of Chinese nursing journals.**

| Practical application of the COREQ/SRQR by journal editors | Number (%) of N = 24 |
|---|---|
| 1. Whether authors are required to comply with the COREQ/SRQR by journal | N = 24 |
| Requested strongly | 0 |
| Advice/recommendation | 8(33.33%,8/24) |
| No requirement at all | 16(66.67%,16/24) |
| 2. Whether editors are required to incorporate the COREQ/SRQR in editorial processes by journal | N = 24 |
| Requested strongly | 0 |
| Advice/recommendation | 7(29.17%,7/24) |
| No requirement at all | 17(70.83%,17/24) |
| 3. Whether peer reviewers are required to incorporate the COREQ/SRQR in peer review processes by journal | N = 24 |
| Requested strongly | 0 |
| Advice/recommendation | 8(33.33%,8/24) |
| No requirement at all | 16(66.67%,16/24) |
| **Plans for the introduction of the COREQ/SRQR in the future** | Nᵃ = 22 |
| 1. Editors who thought it necessary to introduce the COREQ/SRQR into the IFA | 15(68.18%,15/22) |
| 2. Had the intention to introduce the COREQ/SRQR into the IFA of their journals in the near future | 2(13.33%,2/15) |
| 3. Had a specific plan to do so | 0 |

N$^a$ represents nursing journals that have not yet introduced the COREQ/ SRQR into the IFA.

(50%, 12/24); other factors (29.17%, 7/24) including the pioneering role of high-level Chinese nursing journals (12.5%, 3/24) and the widespread application of qualitative research reporting guidelines among Chinese nursing researchers (4.17%, 1/24).

In addition, the editors who completed the survey named several main obstacles to the introduction of the COREQ/ SRQR in the IFA of Chinese nursing journals: the majority of Chinese nursing journal editors lack awareness of the COREQ/SRQR (75%, 18/24); the concern that the majority of submissions would not be published if required to follow the COREQ/SRQR (66.67%, 16/24); the lack of formal approval by the supervisory departments in charge (54.17%, 13/24); reviewers' lack of awareness of the COREQ/SRQR (25%, 6/24); other factors (20.83%, 5/24) such as the lack of widespread application of the COREQ/SRQR among Chinese nursing researchers (4.17%, 1/24).

## Discussion

### Analysis of the endorsement of the COREQ/SRQR in the IFA of Chinese nursing journals

To the best of our knowledge, this study is the first to investigate the endorsement of the COREQ and SRQR in the IFA of Chinese nursing journals, as well as the awareness of, demand for, and application by Chinese editors of the COREQ and SRQR. The COREQ and SRQR were published in 2007 and 2014, respectively [7,9]. They have been endorsed and recommended by the EQUATOR website.[23,24]. However, the results of our investigation were disappointing. Although all of the Chinese nursing journals published qualitative research, only 6.9% (2/29) mentioned the COREQ and SRQR in their IFA. This proportion is not only lower than endorsements of other biomedical research reporting guidelines (such as CONSORT, PRISMA, and STROBE) in the IFA of international or Chinese journals [14,25–27], but is also lower than endorsements of the COREQ/SRQR in the IFA of Nursing Social Science journals.[17]

**Table 3. Reasons for not introducing the COREQ/SRQR into the IFA of Chinese nursing journals.**

| Reasons for not introducing the COREQ/SRQR in the IFA of Chinese nursing journals | Number (%) of N = 24 |
|---|---|
| 1. The main determining factors for adding the COREQ/SRQR to the IFA of journals | N = 24 |
| Formal approval from the supervisory departments in charge | 16(66.67%,16/24) |
| Proposal and request of the editor-in-chief | 14(58.33%,14/24) |
| Consideration of international practices | 12(50%,12/24) |
| Other | 7(29.17%,7/24) |
| 2. Main obstacles to adding the COREQ/SRQR in the IFA of their journals | N = 24 |
| Lack of knowledge of the COREQ/SRQR among the majority of Chinese nursing editors | 18(75%,18/24) |
| The majority of submissions would not be published if required to follow the COREQ/SRQR | 16(66.67%,16/24) |
| Lack of formal approval by the supervisory departments in charge | 13(54.17%,13/24) |
| Lack of knowledge of the COREQ/SRQR among several reviewers | 6(25%,6/24) |
| Other | 5(20.83%,5/24) |

Combined with the findings of this study, we believe that the possible causes are: 1) Poor awareness of the COREQ and SRQR among editors of Chinese nursing journals, confirmed by the results of the survey. The respondents' awareness of both the COREQ and SRQR was low, and the majority considered that this was the main obstacle to the introduction of qualitative research reporting guidelines into Chinese nursing journals. 2) The developmental dilemma of Chinese non-core nursing journals. Studies have shown that fewer manuscripts are submitted to Chinese non-core medical science and technology journals and many of these may be of poor quality compared with manuscripts submitted to high-impact journals [28]. Biomedical research reporting guidelines often have high requirements for the quality of manuscripts, which may lead to further reductions in the publication of manuscripts submitted to these journals if the journals include such guidelines in their IFA [29]. This is consistent with the findings of our study, in which most of the editors of Chinese nursing journals were concerned that the majority of submissions would not be published if they were required to follow the COREQ/SRQR. 3) Lack of formal approval by the periodical management departments. As early as 2004, the International Association of Medical Journal Editors (ICMJE) established the Uniform Requirements for Manuscripts Submitted to Biomedical Journals, which is aimed at regulating manuscript submissions to biomedical journals [30]. However, at present, the official documents issued by the Chinese periodical management departments have not covered the requirements of manuscript submission [31], and only a few Chinese researchers have previously published the initiative to promote the application of reporting guidelines in medical journals [32]. The results of this study also support this view, as more than half of the editors believed that formal approval by authorities is not only the main determinant for introducing the COREQ/SRQR to the IFA of Chinese nursing journals but also the main obstacle to the inclusion of the guidelines in the IFA of the journals.

## Awareness, demand, and application status of the COREQ/SRQR among editors of Chinese nursing journals

Our survey revealed low awareness of both the COREQ and SRQR among editors of Chinese nursing journals, possibly associated with the lack of prompt introduction of the reporting guidelines into their continuing education and training [33]. However, compared with editors who were unaware of the COREQ and SRQR, those who were aware of the COREQ and/or SRQR were much more likely than those who were not to believe that it was necessary to add the COREQ and/or SRQR to the IFA of nursing journals or to follow the COREQ/SRQR in the process of manuscript submission, editing, and peer review. In addition, among those who were unaware of the standards, half were willing to learn more about the

COREQ and SRQR. To some extent, this also reflects a need for effective publicity and training involving the COREQ and/or SRQR in China, which would be key to improving the quality of qualitative research reports on nursing in the country. For example, the COREQ and/or SRQR could be added to continuing education and training programs for the Chinese journal industry and/or to medical school education.

In addition, our survey revealed that the current status of the practical application of the COREQ/SRQR by editors of Chinese nursing journals is poor. None of the surveyed editors had made it mandatory to follow the COREQ/SRQR in the processes of manuscript submission, editing, and peer review. However, studies have shown that the strict implementation of implement reporting guidelines during the processes of manuscript submission, editing, and peer review significantly affects the transparency of the research to be published, since journal editors are directly responsible for determining the reporting format of submitted manuscripts [19,20,34]. Therefore, this may also be an important reason for the poor reporting quality of qualitative research in Chinese nursing journals [35]. Despite this, there is reason to believe that the lack of application of the COREQ/SRQR does not necessarily mean that Chinese nursing journals do not pay attention to the reporting quality of their published qualitative studies. It may just be that different journals have formulated and established different evaluation criteria for submissions.

## Reflections on the value and limitations of reporting guidelines

This study examined the Consolidated Criteria for Reporting Qualitative Research (COREQ) and the Standards for Reporting Qualitative Research (SRQR), two widely endorsed reporting frameworks in nursing qualitative research. A number of critiques have been leveled against qualitative research checklists: excessive rigidity, a tendency to emphasize minor rather than holistic criteria, the confusion between rigorous reporting and rigorous study conduct, and the inappropriate transposition of criteria from one qualitative tradition to another [9,36–38]. A central concern is that such checklists may be misapplied as mechanistic compliance tools, diverting evaluative focus from the substantive, reflective, and theoretically grounded dimensions of inquiry—such as epistemological depth, ethical sensitivity, and authentic representation of participant voices—toward superficial item completion [37]. More fundamentally, transparent and complete reporting does not equate to robust research design or execution; a report fully adherent to COREQ may nonetheless exhibit fundamental deficiencies in philosophical underpinnings, sampling logic, or analytical rigor [9]. Barbour cautioned that dogmatic adherence to "technical procedures" risks "putting the cart before the horse," whereby methodological formalism eclipses substantive engagement with the research question [38]. We acknowledge that subsequent guidelines, notably the American Psychological Association's Journal Article Reporting Standards for Qualitative Research (JARS-Qual), have sought to accommodate diverse research traditions and foreground holistic principles [37].

Nevertheless, while acknowledging these substantive limitations of standardized reporting frameworks, we contend that the prevailing challenge confronting mainland Chinese nursing scholarship is not overreliance or misapplication of guidelines, but rather pervasive unfamiliarity with internationally recognized transparency instruments. Consequently, promoting awareness and appropriate adoption of the COREQ and the SRQR should be positioned as a necessary and urgent foundational step toward advancing the standardization and internationalization of qualitative nursing research in China—not its terminus. Future educational initiatives must emphasize that these frameworks serve as scaffolds for reporting integrity rather than substitutes for methodological rigor, and should cultivate researchers' critical engagement with the epistemological assumptions underlying each criterion.

## Strengths and limitations of this study

The strengths of the study include the analysis of the reasons for the low levels of awareness and application of the COREQ and SRQR reporting guidelines in Chinese nursing journals. This enabled us to propose relevant suggestions for improvement that would provide a scientific reference and guidance for the endorsement, application, and promotion of the COREQ and SRQR in Chinese nursing journals.

 

This study is subject to several limitations. First, the editorial sample was limited ($N = 24$). Despite exhaustive efforts to contact the target population, this sample size constrains statistical power and precludes inferential analyses. Nevertheless, the study's exploratory aim—characterizing the current landscape rather than testing hypotheses or developing predictive models—is adequately served by the available descriptive data. Second, the IFA were retrieved from official websites and, although subject to periodic revision, remained stable throughout the study period. Third, our focus was restricted to nursing journals published in mainland China. While this delimitation enhances applicability to the publishing platforms most widely accessed by the mainland nursing community, it simultaneously limits generalizability to other Chinese-language jurisdictions. Future research might profitably extend to comparative analyses of nursing journals published in the Hong Kong Special Administrative Region, the Macao Special Administrative Region, and Taiwan, China.

## Conclusion

To sum up, at present, the endorsement rate of the COREQ/SRQR in the IFA of nursing journals and awareness of the COREQ/SRQR by journal editors are low in China. None of the editors who participated in the survey had enforced recommendations for authors to follow the COREQ/SRQR at the manuscript submission stage, the editors to follow them at the manuscript processing and editing stages, and the reviewers to use them at the expert review stage. Therefore, we strongly suggest that national periodical management departments such as the CPA should develop appropriate policies to facilitate the introduction of the COREQ/SRQR in the IFA of nursing journals. In addition, effective dissemination of information and education will promote the understanding and use of these reporting guidelines among nursing journal editors, nursing researchers, and nursing students, to increase the utilization of the guidelines in qualitative studies and thus improve the reporting quality of qualitative research in nursing.

## Supporting information

**S1 File. Basic information of the Chinese nursing journals included in this study.**
(DOCX)

**S1 Dataset. Questionnaire dataset.**
(XLSX)

**S1 Checklist. List of research reports.**
(DOCX)

## Acknowledgments

We thank all peer reviewers who participated in the review. We thank MJEditor (www.mjeditor.com) for linguistic assistance during the preparation of this manuscript.

## Author contributions

**Conceptualization:** Li Zhao, Jian Zeng, Kaiyan Hu, bin ma.

**Data curation:** Li Zhao, Jian Zeng, Kaiyan Hu.

**Formal analysis:** Li Zhao, Jian Zeng, Kaiyan Hu.

**Funding acquisition:** Li Zhao, Jian Zeng, bin ma.

**Investigation:** Li Zhao, Jian Zeng, Kaiyan Hu.

**Methodology:** Li Zhao, Jian Zeng, Kaiyan Hu, bin ma.

**Project administration:** Li Zhao, bin ma.

**Resources:** bin ma.

**Supervision:** bin ma.

**Validation:** Li Zhao, Kaiyan Hu, bin ma.

**Visualization:** Li Zhao, Jian Zeng, Kaiyan Hu.

**Writing – original draft:** Li Zhao.

**Writing – review & editing:** bin ma.

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
