## [Decision Letter · Decision Letter 0]

20 Nov 2025

Dear Dr. ma,

Thank you for submitting your manuscript to PLOS ONE. After careful consideration, we feel that it has merit but does not fully meet PLOS ONE’s publication criteria as it currently stands. Therefore, we invite you to submit a revised version of the manuscript that addresses the points raised during the review process.

We look forward to receiving your revised manuscript.

Kind regards,

Alejandro Botero Carvajal, Ph.D

Academic Editor

PLOS ONE

Additional Editor Comments (if provided):

Reviewers' comments:

Reviewer's Responses to Questions

**Comments to the Author**

1. Is the manuscript technically sound, and do the data support the conclusions?

Reviewer #1: No

2. Has the statistical analysis been performed appropriately and rigorously?

Reviewer #1: N/A

3. Have the authors made all data underlying the findings in their manuscript fully available?

Reviewer #1: Yes

4. Is the manuscript presented in an intelligible fashion and written in standard English?

Reviewer #1: Yes

Reviewer #1: General remarks

Objectives:

1. To investigate the endorsement of the 32-item Consolidated Criteria for Reporting Qualitative (COREQ) and Standards for Reporting Qualitative Research (SRQR) in the instructions for authors of Chinese nursing journals.

2. To assess Chinese nursing journals’ editor’s awareness of and demand for the COREQ checklist and SRQR standards, as well as their implementation at different stages of the publication process, including manuscript submission, editing, and peer review

This article represents an interesting study on the quality criteria applied by Chinese nursing journal editors. However, its scope is limited, and a number of points relating to the chosen method should be clarified before publication.

For example, none of the journals included in this study is indexed in Medline and the only journals of nursing published in Chinese speaking countries and indexed in Medline were excluded.

Context:

The COREQ was developed specifically for application to studies using interviews and focus groups, so for these study designs, authors should adhere to the 32-item.

The use of COREQ checklist is appropriate since it improves quality of reporting of original qualitative studies (de Jong et al, BMC Medical Research Methodology (2021) 21:184).

For other qualitative designs, such as case series, ethnography, or narrative research, COREQ does not apply. The authors of SRQR took a more flexible approach attempting to provide a guideline relevant to a broad range of study types, such that SRQR can be applied to any qualitative study design, including focus groups and interviews.

Given the variety of study types and methodological traditions represented within qualitative research, the SRQR and COREQ guidelines best serve their intended aim of improving qualitative study reporting when used as reminders for reporting rather than prescriptive requirements (Dossett et al., JAMA Surg 2021).

Major points

Authors mention that “Walsh et al. recently found that only 28% of the 115 nursing journals surveyed mentioned the COREQ checklist in their IFA and that the reporting quality of qualitative studies published in these 115 nursing journals was inadequate. (Ref 1)]”. Therefore, what is the added value of this study investigating “only” Chinese nursing journals”?

Inclusion criteria: “Nursing journals in the national press and publication database”.

What National Press means? Which databases (this is mentioned in the result section but should be in the method section)? Were only journals publishing in Chinese included?

Is “The Journal of International Nursing Science” different from the “International Journal of Nursing Sciences” (ScienceDirect)?

Is the Journal of Nursing Management in China different from the Journal of Nursing Management (Wiley)

Exclusion criteria: journals from Chinese speaking countries such as those from Hong Kong (e.g. The Hong Kong nursing journal), Macao and Taiwan (e.g. J Nurs Res, or The journal of nursing) were excluded because their inclusion “may have had an impact on the results of the study”. This should be developed.

Investigation of periodical editors: it seams that questions related only to COREQ and SRQR. Did the use of other quality reporting guidelines was investigated?

Results

The number of editors who were aware of the COREQ and/or SRQR is not clear : 11 or 12. It seems that the correct formulation (line 174) should be “Of the 12 editors (50%, 12/24) who knew about the COREQ or the SRQR or both …”

Line 182 “Six of the 12 editors (50%, 6/12) who were not aware of the COREQ/SRQR”: the COREQ or the SRQR or both

Discussion

Authors seem to consider that CORQ and SRQR are the exclusive way to improve quality of reporting, and that their use would ensure that quality.

However, those guidelines (or criteria) have limitations and therefore have faced ongoing critiques for being overly rigid, forcing attention to minor rather than holistic criteria,

conflating reporting rigor with study conduct rigor, and misapplying criteria and techniques from one qualitative tradition to another (doi:10.1093/fampra/cms041; doi:10.1037/amp0000151; doi:10.1097/ACM.0000000000000388; doi:10.1136/bmj.322.7294.1115).

Furthermore, other “guidelines” can be considered. For example, the Journal Article Reporting Standards for Qualitative Research (JARS-Qual Standards), published in 2018, that claims to goes farther than earlier efforts to address the limitations of qualitative reporting guidelines. It was developed by a task force of the American Psychological Association.

Finally, the use of such guidelines can be questioned. In an article is based on a presentation to the British Sociological Association’s Regional Medical Sociology Group in London in March, 2000, Rosaline Barbour stated that reducing qualitative research to a list of technical procedures (such as purposive sampling, grounded theory, multiple coding, triangulation, and respondent validation) was overly prescriptive, resulted in “the tail wagging the dog” and that none of these “technical fixes” in itself conferred rigour (Barbour, BMJ 2001).

Those points should be discussed.

Minor points

Table 1: SROR instead of SRQR

Line 344 : “recom-mendations” remove “-“

.

Reviewer #1: No

---

## [Author Response · Author response to Decision Letter 1]

15 Mar 2026

Dear editors and Reviewers :

Thank you for giving us the opportunity to submit a revised draft of the manuscrip“Endorsement of the 32-item Consolidated Criteria for Reporting Qualitative (COREQ) and Standards for Reporting Qualitative Research (SRQR) by Chinese journals of nursing: A survey of editors and review of journal instructions for authors” (PONE-D-25-46330) for publication in the Journal of PLoS One.We appreciate the time and effort that you and the reviewers dedicated to providing feedback on our manuscript. We are grateful for the thoughtful comments on the manuscript.We have incorporated changes to reflect most of the suggestions made by the reviewers. We have highlighted the changes within the revised manuscript. Modifications in the revised manuscript are distinctly indicated using the Track Changes feature.

Here is a point-by-point response to the reviewers’ comments.

Response to Reviewer 1

Comment 1: Is the manuscript technically sound, and do the data support the conclusions?

Reviewer #1: No

Response:We appreciate the reviewers' emphasis on the importance of methodological rigor.This study employs a cross-sectional survey design. Given our small sample size (N=24), which would significantly compromise the power and stability of statistical inferences, and considering that the primary objective is to describe the current situation rather than test hypotheses, we ultimately opted for robust descriptive analysis. This approach ensures that conclusions are correctly drawn from the data without exceeding its analytical limits. It is an appropriate and common practice in small-scale, census-based studies of this nature. When reporting percentages, we adhere to clear reporting standards by providing both the numerator and denominator (e.g., 45.8% (11/24)) to avoid ambiguity.In response,we have conducted comprehensive and meticulous revisions to the manuscript's Methods and Limitations sections, to enhance its technical reliability. Please refer to the revised manuscript: Page 6, lines 160–162; Page 14, lines 312–316.

Comment 2: Has the statistical analysis been performed appropriately and rigorously?

Reviewer #1: N/A

Response: Thanks for pointing this out. We have thoroughly and meticulously revised the Methods and Limitations section of the manuscript. Please refer to the revised manuscript: Page 6, lines 160–162; Page 14, lines 312–316.

Comment 3: Have the authors made all data underlying the findings in their manuscript fully available?

Reviewer #1: Yes

Response: We agree with this and all relevant data are within the manuscript and its Supporting Information files.

Comment 4: Is the manuscript presented in an intelligible fashion and written in standard English?

Reviewer #1: Yes

Response: We have conducted a thorough language verification of the manuscript and identified a translation error, which has been corrected in the revised version (see Line 139, Page 5). Additionally, the references have also been reviewed to ensure correct and consistent citation style and formatting in the revised manuscript.

Comment 5: This article represents an interesting study on the quality criteria applied by Chinese nursing journal editors. However, its scope is limited, and a number of points relating to the chosen method should be clarified before publication.

For example, none of the journals included in this study is indexed in Medline and the only journals of nursing published in Chinese speaking countries and indexed in Medline were excluded.

Response: Thank you for recognizing the value of the study and for your insightful comments on the "Materials and Methods" section.The journals included in this study are currently not indexed in Medline. We have revised the "Materials and Methods" section accordingly; please refer to lines 109–129 on pages 4–5 of the revised manuscript.This study focuses on Chinese nursing journals published in mainland China and primarily targeting the mainland Chinese nursing academic community. This focus enables the research to provide a deeper and more homogeneous reflection of the current status of recognition and application of the COREQ and SRQR guidelines for qualitative research reports within the mainstream academic publishing platforms in mainland China.This is essential for the development of a targeted localized outreach strategy. Of course, we fully agree with the characteristics of the scope that you point out, as we have clearly indicated in the "Limitations" paragraph:“Our focus was restricted to nursing journals published in mainland China. While this delimitation enhances applicability to the publishing platforms most widely accessed by the mainland nursing community, it simultaneously limits generalizability to other Chinese-language jurisdictions. Future research might profitably extend to comparative analyses of nursing journals published in the Hong Kong Special Administrative Region, the Macao Special Administrative Region, and Taiwan, China.”See Revised Manuscript, page 14, lines 317-322.

Comment 6: Authors mention that “Walsh et al. recently found that only 28% of the 115 nursing journals surveyed mentioned the COREQ checklist in their IFA and that the reporting quality of qualitative studies published in these 115 nursing journals was inadequate. (Ref 1)]”. Therefore, what is the added value of this study investigating “only” Chinese nursing journals”?

Response: Thank you for sharing this insightful perspective. Our research does not replicate the findings of Walsh et al. Specifically, the added value of this study lies primarily in the following three aspects:(1) In recent years, there has been a significant increase in the number of qualitative studies in the field of nursing in China (Ref18), and the quality of reporting has been poor (Ref33). However, the implementation of academic publishing norms is deeply influenced by local research ecosystems, editorial practice traditions, and policy guidance.Therefore,while findings at the international level are informative, they cannot directly map or replace the understanding of the local Chinese context.This study adds to the systematic evidence addressing the adoption of COREQ/SRQR by nursing journals in mainland China.(2) Our study goes a step further than Walsh et al. by using a hybrid design:The IFA and applications of the COREQ and SRQR were assessed. The editors of the journals were asked about their awareness of and demand for the COREQ checklist and SRQR standards, as well as their implementation at different stages of the publication process, including manuscript submission, editing, and peer review.We not only quantified the formal adoption rate of the COREQ/SRQR guidelines for qualitative research reports in Chinese nursing journals, but also delved deeper into the nursing editorial community's awareness and needs regarding these guidelines. We examined their actual application throughout the publishing process (submission, editing, and peer review), as well as the primary factors driving or hindering the incorporation of the COREQ checklist or SRQR standards into journal IFAs.Walsh et al. did not touch on this dimension, and our study informs this.(3)Practical implications: The practical implication of this study is to drive change. Based on these findings, our recommendations are not generalized.For example, we emphasize the urgent need for local authoritative bodies such as the China Periodical Association and the Chinese Nursing Association to spearhead the development of promotional measures.Rooted in a deep understanding of the academic ecology of mainland China, these recommendations have a stronger operational and policy reference value.

Comment 7: Inclusion criteria: “Nursing journals in the national press and publication database”.

What National Press means? Which databases (this is mentioned in the result section but should be in the method section)? Were only journals publishing in Chinese included?

Is “The Journal of International Nursing Science” different from the “International Journal of Nursing Sciences” (ScienceDirect)?

Is the Journal of Nursing Management in China different from the Journal of Nursing Management (Wiley)

Response: Thank you very much for pointing out these ambiguities. We have integrated the description of the database from the "Results" section into the "Methods" section. Please refer to lines 109–129 on pages 4–5 of the revised manuscript.The National Press and Publication Administration (NPPA) serves as the highest administrative authority for journal registration and management in mainland China. Its official list constitutes the authoritative basis for confirming a journal's legal publication status within mainland China. Therefore, we first obtained all registered "nursing" journals from the NPPA's official website to form an initial journal pool.The journals included in this study were all nursing journals published in mainland China. Publication languages: 2 were in English, 1 was in Chinese/English, and the rest were in Chinese.Clarification regarding specific journals: (1) The term "the Journal of International Nursing Science" in the original manuscript was a translation error. The correct English name of the journal is “International Journal of Nursing Sciences”(ScienceDirect;ISSN 2352-0132); (2) The term "the Journal of Nursing Management in China" in the original manuscript was a translation error. The correct English name of the journal is "Chinese Nursing Management”(ISSN 1672-1756)".We sincerely apologize for this oversight. The revisions have been incorporated into the revised version. Please refer to page 7, line 176, and page 7, line 177 of the revised version.To ensure accuracy, we have double-checked all included journals’ Journal Name、Database、Language、ISSN and Website in S1file .We made one revision: the journal of “Nursing of Integrated Traditional Chinese and Western Medicine” was renamed to "Nursing of Integrated Traditional Chinese and Western Medicine" in 2022, and its website has been updated to: https://www.nursing-tcwm.com.

Comment 8: Exclusion criteria: journals from Chinese speaking countries such as those from Hong Kong (e.g. The Hong Kong nursing journal), Macao and Taiwan (e.g. J Nurs Res, or The journal of nursing) were excluded because their inclusion “may have had an impact on the results of the study”. This should be developed.

Response: We fully agree with your suggestion. We have incorporated it into the revised manuscript, as shown on pages 4–5, lines 109–129.

Comment 9: Investigation of periodical editors: it seams that questions related only to COREQ and SRQR. Did the use of other quality reporting guidelines was investigated?

Response: Thank you for raising this important issue. We fully agree that it is also valuable to investigate the adoption of other types of reporting guidelines (e.g., CONSORT, STROBE, etc.). We did limit the survey to the COREQ and SRQR, based on two main considerations:(1)Research Objective: The core objective of this study is to evaluate the current adoption status of reporting standards for qualitative research in Chinese nursing journals. COREQ and SRQR are the two most widely used international reporting standards officially recommended by the EQUATOR Network, specifically designed for qualitative research. Investigating other types of guidelines (e.g., CONSORT for randomized controlled trials, STROBE for observational studies), while also valuable, would detract from our core objectives.(2)Methodology: Simultaneously inquiring in depth about the awareness and application of multiple different types of reporting guidelines would result in a lengthy questionnaire, diffuse focus, and could significantly reduce both the response rate and the quality of responses from editors. By focusing on these two specific guidelines, we are able to conduct a more in-depth investigation. For example, we can further examine the barriers preventing COREQ and SRQR from being integrated into the standard workflow of nursing journals in mainland China.

Comment 10: Results：

The number of editors who were aware of the COREQ and/or SRQR is not clear : 11 or 12. It seems that the correct formulation (line 174) should be “Of the 12 editors (50%, 12/24) who knew about the COREQ or the SRQR or both …”

Line 182 “Six of the 12 editors (50%, 6/12) who were not aware of the COREQ/SRQR”: the COREQ or the SRQR or both

Response: Thank you for pointing this out. We agree with you and have incorporated your suggestion. The revisions can be found at line 191 on page 7 and line 200 on page 8.

Comment 11: Discussion：

Authors seem to consider that CORQ and SRQR are the exclusive way to improve quality of reporting, and that their use would ensure that quality.

However, those guidelines (or criteria) have limitations and therefore have faced ongoing critiques for being overly rigid, forcing attention to minor rather than holistic criteria,conflating reporting rigor with study conduct rigor, and misapplying criteria and techniques from one qualitative tradition to another (doi:10.1093/fampra/cms041; doi:10.1037/amp0000151;doi:10.1097/ACM.0000000000000388;doi:10.1136/bmj.322.7294.1115).

Furthermore, other “guidelines” can be considered. For example, the Journal Article Reporting Standards for Qualitative Research (JARS-Qual Standards), published in 2018, that claims to goes farther than earlier efforts to address the limitations of qualitative reporting guidelines. It was developed by a task force of the American Psychological Association.

Finally, the use of such guidelines can be questioned. In an article is based on a presentation to the British Sociological Association’s Regional Medical Sociology Group in London in March, 2000, Rosaline Barbour stated that reducing qualitative research to a list of technical procedures (such as purposive sampling, grounded theory, multiple coding, triangulation, and respondent validation) was overly prescriptive, resulted in “the tail wagging the dog” and that none of these “technical fixes” in itself conferred rigour (Barbour, BMJ 2001).

Those points should be discussed.

Response: We sincerely thank you for your profound and important comments.Your feedback has helped us achieve a crucial elevation: transforming this study from an investigation into "specific tool usage" into a more reflective discussion about "how to responsibly introduce and utilize reporting tools within a specific academic ecosystem to enhance overall research quality."We believe that these revisions have greatly enhanced the academic depth and rigor of the paper.For specific revisions, please refer to lines 295–321 on page 13-14 of the revised manuscript.

Comment 12: Minor points：

Table 1: SROR instead of SRQR

Line 344 : “recom-mendations” remove “-“

Response: We appreciate the reviewer’s careful attention to these details. The relevant sections have been revised in response to the minor points, as follows: line 188 (page 8) and line 351 (page 15).

---

## [Decision Letter · Decision Letter 1]

25 Mar 2026

Endorsement of the 32-item Consolidated Criteria for Reporting Qualitative (COREQ) and Standards for Reporting Qualitative Research (SRQR) by Chinese journals of nursing: A survey of editors and review of journal instructions for authors

PONE-D-25-46330R1

Dear Dr. ma,

We’re pleased to inform you that your manuscript has been judged scientifically suitable for publication and will be formally accepted for publication once it meets all outstanding technical requirements.

Kind regards,

Alejandro Botero Carvajal, Ph.D

Academic Editor

PLOS One

Additional Editor Comments (optional):

Reviewers' comments:

Reviewer's Responses to Questions

**Comments to the Author**

Reviewer #1: All comments have been addressed

2. Is the manuscript technically sound, and do the data support the conclusions?

Reviewer #1: Yes

3. Has the statistical analysis been performed appropriately and rigorously?

Reviewer #1: N/A

4. Have the authors made all data underlying the findings in their manuscript fully available?

Reviewer #1: Yes

5. Is the manuscript presented in an intelligible fashion and written in standard English?

Reviewer #1: Yes

Reviewer #1: (No Response)

.

Reviewer #1: **Yes:** Jean-Francois GEHANNOJean-Francois GEHANNOJean-Francois GEHANNOJean-Francois GEHANNO

---

## [Editor Report · Acceptance letter]

PONE-D-25-46330R1

PLOS One

Dear Dr. ma,

I'm pleased to inform you that your manuscript has been deemed suitable for publication in PLOS One. Congratulations! Your manuscript is now being handed over to our production team.

Kind regards,

on behalf of

Dr. Alejandro Botero Carvajal

Academic Editor

PLOS One